# Molecular Characterization of the Acyl-CoA-Binding Protein Genes Reveals Their Significant Roles in Oil Accumulation and Abiotic Stress Response in Cotton

**DOI:** 10.3390/genes14040859

**Published:** 2023-04-01

**Authors:** Yizhen Chen, Mingchuan Fu, Hao Li, Liguo Wang, Renzhong Liu, Zhanji Liu

**Affiliations:** Key Laboratory of Cotton Breeding and Cultivation in Huang-Huai-Hai Plain, Ministry of Agriculture and Rural Affairs, Institute of Industrial Crops Shandong Academy of Agricultural Sciences, Jinan 250100, China

**Keywords:** cotton, *ACBP* gene family, expression analysis, salt and drought stress

## Abstract

Members of the acyl-CoA-binding protein (ACBP) gene family play vital roles in diverse processes related to lipid metabolism, growth and development, and environmental response. Plant *ACBP* genes have been well-studied in a variety of species including Arabidopsis, soybean, rice and maize. However, the identification and functions of *ACBP* genes in cotton remain to be elucidated. In this study, a total of 11 *GaACBP*, 12 *GrACBP*, 20 *GbACBP*, and 19 *GhACBP* genes were identified in the genomes of *Gossypium arboreum*, *Gossypium raimondii*, *Gossypium babardense,* and *Gossypium hirsutum*, respectively, and grouped into four clades. Forty-nine duplicated gene pairs were identified in *Gossypium ACBP* genes, and almost all of which have undergone purifying selection during the long evolutionary process. In addition, expression analyses showed that most of the *GhACBP* genes were highly expressed in the developing embryos. Furthermore, *GhACBP1* and *GhACBP2* were induced by salt and drought stress based on a real-time quantitative PCR (RT-qPCR) assay, indicating that these genes may play an important role in salt- and drought-stress tolerance. This study will provide a basic resource for further functional analysis of the *ACBP* gene family in cotton.

## 1. Introduction

Plant acyl-CoA-binding proteins (ACBPs) comprise a highly conserved family that bind to a variety of acyl-coenzyme A esters with high specificity and affinity through an acyl-CoA-binding (ACB) domain [1]. Based on molecular weight, subcellular localizations, and the presence of a kelch motif or ankyrin repeats domain, plant ACBPs are often categorized into four classes, namely small ACBPs, ankyrin-ACBPs, large ACBPs, and kelch-ACBPs [2,3]. The 10 kDa protein encoded by *BnACBP*, the first plant *ACBP* gene, was discovered in *Brassica napus* [4]. Subsequent research showed that BnACBP can bind both C16:0-CoA and C18:1-CoA esters [5]. Many *ACBP* genes have now been found in a variety of plant species, but the number of *ACBP* genes in different species varies considerably. Six *ACBP* genes have been found in *Arabidopsis thaliana* [6], 6 in *Oryza sativa* [7], 9 in *Zea mays* [8], 11 in *Glycine max* [2], 15 in *Arachis hypogaea* [3], and 19 in *B. napus* [9]. However, little information on the *ACBP* gene family has been reported in cotton to date.

Extensive studies show that plant *ACBP* genes are involved in a wide range of cellular processes including modulation of lipid metabolism, regulation of gene expression, and regulation of plant growth and development [1,2,3]. For example, overexpression of *BnACBP* altered the fatty acid composition in Arabidopsis developing seeds [10]. *AtACBP1* and *AtACBP2* have been shown to be involved in lipid transfer. Simultaneous mutation of *AtACBP1* and *AtACBP2* results in embryo lethality [11]. Furthermore, increasing evidences suggest that plant *ACBP* genes can be essential for specific environmental conditions such as drought, high salinity or low temperature. *AtACBP2* is induced by drought and ABA. Overexpression of *AtACBP2* confers transgenic Arabidopsis enhanced drought tolerance [12].

As a major cash crop, cotton not only provides natural fiber for the textile industry, but is also an important source of vegetable oil and protein. The allotetraploid *G*. *hirsutum* (2*n* = 4*x* = 52, AD_1_) and *G. barbadense* (2*n* = 4*x* = 52, AD_2_), which are cultivated worldwide for their economic value, evolved from an occasional hybridization between the A- and D-genome ancestors, followed by chromosome doubling. The diploid *G. arboreum* (2*n* = 2*x* = 26, A_2_) and *G. raimondii* (2*n* = 2*x* = 26, D_5_) are closely related to the A- and D-genome ancestors, respectively [13]. Although some *ACBP* genes have been well-characterized in *A. thaliana*, particularly with regard to the regulation of lipid metabolism and stress tolerance [11,12], the function of *ACBP* genes in cotton has not been identified to date. In this study, we performed a genome-wide characterization of the *ACBP* gene family in four cotton species. The phylogenetic relationships, gene structure, conserved motifs, gene duplication, *cis*-acting regulatory elements, and expression patterns in different tissues and in response to drought and high salinity were comprehensively analyzed. Our results will provide the basic resource for further functional analysis of the *ACBP* gene family.

## 2. Materials and Methods

### 2.1. Identification of Cotton ACBP Genes

The genome data of *G. arboreum* (CRI, v1.0) [13], *G. raimondii* (JGI, v2.0) [14], *G. barbadense* (HAU, v2.0) [15], and *G. hirsutum* (HAU, v1.1) [15] were downloaded from the CottonFGD database (https://cottonfgd.net/, (accessed on 17 September 2022)) [16]. The Hidden Markov Model (HMM) profile of the ACBP domain (PF00887) retrieved from the InterPro database (https://www.ebi.ac.uk/interpro/entry/pfam/PF00887/, (accessed on 17 September 2022)) [17] was used as a query to search against the protein sequences of the four *Gossypium* species with an e-value < 1 × 10^−10^. The resulting hits were further validated in the presence of ACBP domain by the SMART tool (http://smart.embl.de/, (accessed on 19 September 2022)) [18]. The molecular weight (MW) and theoretical isoelectric point (pI) of each ACBP protein were computed using Compute pI/Mw tool (https://web.expasy.org/compute_pi/, (accessed on 19 September 2022)).

### 2.2. Phylogenic, Gene Structure, and Conserved Motif Analyses

The ACBP domain sequences of *ACBP* genes identified in four *Gossypium* genomes and previously reported in Arabidopsis and rice [7] were used for phylogenetic analysis. A maximum likelihood (ML) tree was constructed by MEGA 11 [19] using 1000 bootstrap replicates. The substitution model was Jones–Taylor–Thornton (JTT). The exon–intron structures were retrieved from the GFF files of four *Gossypium* genomes. The ten conserved motifs were identified using MEME online software (http://meme-suite.org/tools/meme, (accessed on 20 September 2022)) [20]. The phylogenetic tree, gene structure, and conserved motifs were illustrated by TBtools [21].

### 2.3. Chromosomal Mapping and Gene Duplication Analyses

The chromosomal location information of *ACBP* genes was extracted from GFF gene annotation files and then visualized with the MapChart software [22]. The duplication pattern for each *ACBP* gene was analyzed by the method reported in the pear sugar transporter family genes [23]. For example, the 70,199 genes from the *G. hirsutum* genome were aligned using an all-vs-all local BLAST search with an e-value < 1 × 10^−10^. MCScanX software [24] was used to analyze the BLAST outputs and detect gene synteny and collinearity. Collinear gene pairs on two segmental loci were considered segmental duplication, whereas tandem duplications were characterized as adjacent homologous genes on a single chromosome without an intervening gene.

### 2.4. Promoter Analysis for Cis-Acting Regulatory Elements

The promoter sequences of *GhACBP* genes (1500 bp upstream) were retrieved from *G. hirsutum* genome sequences [15]. The *cis*-acting regulatory elements in the promoter region of each *GhACBP* gene were predicted using the PlantCARE online tool (https://bioinformatics.psb.ugent.be/webtools/plantcare/html/ (accessed on 23 September 2022)) [25].

### 2.5. Expression Profile Analysis of GhACBP Genes

The expression profiles of *GhACBP* genes in different tissues were determined based on public transcriptome data of genotypes TM-1, 11-0509 and Emian22 [26,27,28]. Transcript levels were calculated as fragments per kilobase of exon model per million mapped fragments (FPKM). The heat maps were visualized using the TBtools software (v1.098769) [21]. Furthermore, the ankyrin-ACBPs clade genes were selected to analyze their response to abiotic stress by RT-qPCR.

Healthy seeds of the *G. hirsutum* cultivar Lumian451 were grown in commercial soil at 28 °C with a photoperiod of 16 h light/8 h dark. Two-week-old seedlings were gently uprooted, rinsed, and cultivated in Hoagland’s solution for two days. These seedlings were then randomly divided into three groups for 200 mM NaCl, 15% PEG6000 and 100 μM ABA, respectively. Total RNA was isolated from the leaves. RT-qPCR was carried out as described by Chen et al. (2022) [29]. The experiments were biologically repeated three times, and the relative expression levels of the *GhACBP* genes were calculated based on the 2^-ΔΔCt^ method [30]. The RT-qPCR primers are listed in Appendix A.

## 3. Results

### 3.1. Identification and Phylogenetic Analysis of Gossypium ACBP Genes

To mine *ACBP* genes in cotton, a genome-wide identification was carried out using HMMER searches with the ACBP domain (PF00887) as a query in the protein database of four *Gossypium* species, *G. arboreum*, *G. raimondii*, *G. barbadense*, and *G. hirsutum*. A total of 62 non-redundant *ACBP* genes were identified, including 11, 12, 20, and 19 in *G. arboreum*, *G. raimondii*, *G. barbadense*, and *G. hirsutum*, respectively. Based on previous reports in Arabidopsis and rice, the *ACBP* genes of *G. arboreum*, *G. raimondii*, *G. barbadense*, and *G. hirsutum* were named *GaACBP1* to *GaACBP11*, *GrACBP1* to *GrACBP12*, *GbACBP1* to *GbACBP20*, and *GhACBP1* to *GhACBP19*, respectively. The detailed information of the *Gossypium ACBP* genes, including gene locus, chromosome location, exon number, sequence length, MW, and pI are listed in Table 1. The amino acid length of the *Gossypium* ACBP proteins ranged from 85 (GrACBP11) to 679 (GbACBP9 and GbACBP10), with corresponding MWs varying from 9.59 (GrACBP11) to 74.61 (GbACBP9) kDa. Their pI ranged from 3.98 (GrACBP2) to 9.48 (GbACBP16) (Table 1).

To reveal the evolutionary relationship of the *ACBP* genes in cotton, a total of 74 *ACBP* genes, including 62 *Gossypium ACBP* genes, 6 *AtACBP* genes from *A*. *thaliana* and 6 *OsACBP* genes from *O*. *sativa*, were used to construct an ML phylogenetic tree using the MEGA11 software [19]. As shown in Figure 1 and Appendix A, all *ACBP* genes were classified into four distinct clades, namely small ACBPs, large ACBPs, ankyrin-ACBPs, and kelch-ACBPs (Appendix A). The number of *Gossypium ACBP* genes in the four clades was 30, 7, 6, and 19, respectively. Each *Gossypium* gene of the kelch-ACBPs clade contains one ACBP domain in the N-terminus and three kelch domains in the C-terminus. In addition to the ACBP domain, all members of the ankyrin-ACBPs clade contain a C-terminal ankyrin repeat. The *ACBP* genes of small ACBPs and large ACBP clades contain only one ACBP domain, but they differ in the location of the ACBP domain (Appendix A).

### 3.2. Gene Structure and Conserved Motif Analysis of Cotton ACBP Genes

The number of exons in *Gossypium ACBP* genes ranged from 3 to 19 (Figure 1). We found that *Gossypium ACBP* genes in the same clade had similar gene structures. Specifically, all members of the ankyrin-ACBPs and large ACBPs clades had six and five exons, respectively. In addition, most genes of the small ACBPs clade had 4 exons, and most members of the kelch-ACBPs clade had 18 exons.

We analyzed the motif distribution of the 62 *Gossypium* ACBP proteins and found that each *Gossypium* ACBP protein had different conserved motifs ranging from 1 to 7 (Figure 1). Among the 10 conserved motifs, motif 2 is present in 42 *Gossypium* ACBP proteins, followed by motif 5 in 37 *Gossypium* ACBP proteins. Notably, members of different clades showed unique distribution patterns of conserved motifs. Specifically, all *Gossypium* ACBP proteins in large ACBPs clade contained motifs 2 and 9, members in ankyrin-ACBPs clade all had motifs 2, 6, and 7, and all proteins except GhACBP8 (missing motif 2) in kelch-ACBPs clade had motifs 1, 2, 3, 4, 5, and 8. In the small ACBPs clade, 18 *Gossypium* ACBP proteins had only motif 5, and the remaining 12 members except GhACBP19 (missing motif 9) contained motifs 2, 9, and 10 (Figure 1). In conclusion, *Gossypium ACBP* genes in the same clade generally have similar gene structures and motif distribution patterns.

### 3.3. Genomic Localization and Gene Duplication Analysis of Cotton ACBP Genes

According to the sequenced genome data, the 62 *Gossypium ACBP* genes were physically anchored to 35 specific chromosomes in four *Gossypium* species (Figure 2). Specifically, a total of 11 *GaACBP* genes were mapped to six chromosomes of *G. arboreum*. Chromosome 11 contained five *GaACBP* genes, chromosome 13 contained two *GaACBP* genes, and chromosomes 6, 7, 9, and 12 contained only one *GaACBP* gene each. Similarly, the 12 *GrACBP* genes were located on 6 chromosomes, with five *GrACBP* genes on chromosome 7, three on chromosome 13, and one each on chromosomes 1, 6, 8, and 10. Of the 19 *GhACBP* genes identified in *G. hirsutum,* 10 members were anchored to six chromosomes on the A-subgenome, and 9 *GhACBP* genes were mapped to five chromosomes on the D-subgenome. Chromosomes A11 and D11 contained five *GhACBP* genes each, while the remaining nine chromosomes contained only one *GhACBP* gene each. In *G. babardense*, 20 *GbACBP* genes were mapped to 12 chromosomes with 5 *GbACBP* genes on chromosome A11, 4 on chromosome D11, 2 on chromosome D13, and 1 each on chromosomes A06, A07, A09, A12, A13, D06, D07, D09, and D12. In addition, 44 *ACBP* genes (70.97%) were distributed at both ends of chromosomes, such as *GaACBP2* at the top of chromosome Chr06 and *GbACBP1* at the bottom of chromosome A09. In conclusion, the *Gossypium ACBP* genes were unevenly distributed on their chromosomes, with most *ACBP* genes located at both ends of the chromosomes.

To reveal the expansion of *Gossypium ACBP* genes, gene duplication analysis was performed in the four *Gossypium* species using the MCScanX program [24] and the coding sequences of all genes, and the details of duplicated pairs are listed in Table 2. A total of 6, 8, 18, and 17 pairs of duplicated *ACBP* genes were detected in *G. arboreum*, *G. raimondii*, *G. barbadense*, and *G. hirsutum*, respectively. Specifically, 17 pairs of *GhACBP* genes, involving 16 *GhACBP* genes, were segmental duplications, and 2 pairs (*GhACBP12*/*GhACBP13* and *GhACBP17*/*GhACBP18*) were tandem duplications within the *G. hirsutum* genome (Table 2 and Appendix A). Similarly, segmentally duplicated *ACBP* genes were dominant in *G. arboreum*, *G. raimondii*, and *G. barbadense*. These results suggest that segmental duplication played a more important role in the expansion of the *Gossypium ACBP* genes than tandem duplication. Furthermore, all duplication pairs, except *GhACBP3*/*GhACBP4* and *GhACBP12*/*GhACBP17*, had Ka/Ks values less than 1, ranging from 0.194 to 0.801, indicating that the vast majority of *ACBP* genes in *Gossypium* species have been subjected to purifying selection during the long evolutionary process.

### 3.4. Cis-Acting Regulatory Analysis of GhACBP Genes’ Promoters

We identified a number of *cis*-acting regulatory elements from the 1500 bp upstream regions of the *GhACBP* genes. Apart from eukaryotic basal regulatory elements such as TATA-box and CAAT-box, eight and nine regulatory elements related to phytohormone responsiveness and environmental stress responsiveness, respectively, were identified in the 19 *GhACBP* genes (Figure 3). Each *GhACBP* gene contains at least two phytohormone-responsive elements and two stress-responsive elements. In addition, the *cis*-acting elements for ABA response (ABRE), ethylene response (ERE), anoxic response (ARE), and drought response (MYB, MYB-like, and MYC) are present in most *GhACBP* genes, whereas the element for GA response (TATC-box) is present in *GhACBP1*, *GhACBP6,* and *GhACBP12*. Interestingly, the divergence of regulatory elements occurred in all duplicated *GhACBP* genes. For example, only one out of five *cis*-acting elements for hormone response was shared by the *GhACBP1*/*GhACBP2* duplicate pair (Figure 3). These results suggest that *GhACBP* genes may be differentially regulated by different transcription factors.

### 3.5. Expression Pattern Analysis of GhACBP Genes

According to the transcriptomic data of upland cotton genotypes TM-1, 11-0509, and Emian22 [26,27,28], the expression profiles of *GhACBP* genes in different tissues or developmental stages were analyzed and visualized by a heat map (Figure 4). As shown, *GhACBP* genes were differentially expressed in Leaf, Root, Stem, and Ovules. *GhACBP12*, *GhACBP13*, *GhACBP17,* and *GhACBP18* were highly expressed in all tissues. In contrast, *GhACBP10* and *GhACBP15* were not expressed in all detected tissues. Notably, 14 of the 17 expressed *GhACBP* genes had their highest expression levels in the Ovule (1~20 dpa, days post anther), while *GhACBP11*, *GhACBP14,* and *GhACBP19* were highly expressed in the Leaf (Figure 4). In addition, *GhACBP* genes had similar expression patterns in developing embryos of two cotton genotypes 11-0509 and Emian22 with remarkably different seed oil content. In particular, *GhACBP12*, *GhACBP13*, *GhACBP17,* and *GhACBP18* had significantly higher expression levels in embryos at 10 and 20 dpa compared to other *GhACBP* genes (Figure 4).

Previous studies have shown that drought or salt treatment induces expression of the ankyrin-ACBPs clade genes [12,31]. We evaluated the expression patterns of *GhACBP1* and *GhACBP2* after exposure to 15% PEG6000, 200 mM NaCl, and 100 μM ABA, respectively. As shown in Figure 5, the expression levels of *GhACBP1* and *GhACBP2* were significantly altered under drought, salt, and ABA treatments. Furthermore, *GhACBP1* is more sensitive to drought stress than *GhACBP2*. In addition, the expression of *GhACBP2* increased remarkably at 1h after salt treatment, decreased significantly at 3h, then increased at 6h and finally peaked at 12h (Figure 5).

## 4. Discussion

Since the discovery of the first plant *ACBP* gene, *BnACBP*, in 1994, plant *ACBP* genes have been studied for nearly three decades and have been functionally implicated in many physiological processes, such as fatty-acid metabolism, growth and development, and stress tolerance [3]. In this study, we performed a systematic analysis of the cotton *ACBP* genes to investigate their potential functions in oil accumulation and abiotic stress response. A total of 62 *ACBP* genes were identified in the four *Gossypium* genomes, including 19 *GhACBP* genes in *G. hirsutum* (Table 1). Based on phylogenetic analysis, the cotton *ACBP* genes were classified into four distinct clades, namely small ACBPs, ankyrin-ACBPs, large ACBPs and kelch-ACBPs (Figure 1), which is consistent with the results reported in rice [7], maize [8], and soybean [2]. In particular, the small ACBPs and kelch-ACBPs clades have expanded in cotton compared to those in Arabidopsis. For example, the two diploid species, *G. arboreum* and *G. raimondii*, each contain 5 *AtACBP6* paralogs, whereas the two tetraploid species, *G. hirsutum* and *G. barbadense*, each contain 10 *AtACBP6* paralogs (Figure 1). Furthermore, we analyzed the *ACBP* gene duplication in the four cotton genomes and identified 49 duplicated *ACBP* gene pairs, including 43 segmental duplicates and 6 tandem duplicates (Table 2). In addition, 47 of the 49 duplicated gene pairs had undergone purifying selection during evolution based on the Ka/Ks analysis. These results suggest that segmental duplication and purifying selection may have played an important role in the evolution of the *ACBP* gene family in cotton.

Accumulating evidence showed that many *ACBP* genes were found to be involved in lipid metabolism [1,3]. *AtACBP6*, the small *ACBP* gene, expressed in all tissues of Arabidopsis. Ectopic expression of *AtACBP6* altered erucic acid levels in transgenic oilseed rape seeds [32]. *BnACBP6*, the *AtACBP6* ortholog, has been shown to be expressed in all tissues and to a greater extent in developing embryos and flowers [4,33]. Overexpression of *BnACBP6* significantly increased 18:2 and 18:3 levels and decreased 20:1, 16:0, and 18:0 levels in transgenic Arabidopsis seed oil [10]. In this study, the small ACBPs clade contains 10 *GhACBP* genes (*GhACBP10*–*GhACBP19*). Expression analysis shows that *GhACBP12*, *GhACBP13*, *GhACBP17,* and *GhACBP18* are highly expressed in 10 and 20 dpa embryos (Figure 4), suggesting that these four *GhACBP* genes may play a crucial role in seed oil accumulation and relate with the oil contents in cotton. Recently, *ACBP6* (*GhACBP13* in this study) was shown to be highly expressed in developing cotton embryos. Overexpression of *G. barbadense ACBP6* significantly increased oil content in transgenic yeast. In addition, the expression level of *ACBP6* in *G. barbadense* acc. 3–79 (33.79% seed oil content), was remarkably higher than that in *G. hirsutum* cv. Emian22 (24.97%) during almost the entire seed development process, suggesting that *ACBP6* may play a decisive role in the accumulation of high oil content [28]. Based on amino acid alignment, GhACBP13 shares 79.55% and 80.90% sequence identity with the AtACBP6 and BnACBP6, respectively, indicating that *GhACBP13*, like *AtACBP6* and *BnACBP6*, may influence the fatty-acid composition in cotton seeds.

Plant *ACBP* genes, particularly the ankyrin-ACBPs clade, have been implicated in several stress responses, including salt stress and drought stress [7,8,12]. In Arabidopsis, the ankyrin-ACBPs clade consists of two members, *AtACBP1* and *AtACBP2*. *AtACBP1* was induced by NaCl and mannitol treatments. Transgenic Arabidopsis plants overexpressing *AtACBP1* showed reduced tolerance to salt and mannitol treatments, whereas the *acbp1* mutant exhibited the opposite phenotype [31]. *AtACBP2* was induced by drought and ABA. Overexpression of *AtACBP2* increased drought tolerance and enhanced sensitivity to ABA treatment in Arabidopsis [12]. In soybean, *GmACBP3* and *GmACBP4* belong to the ankyrin-ACBPs clade and share 98.02% amino acid sequence identity [2]. Under salt treatment, the transcript levels of *GmACBP3, GmACBP4,* and their alternatively spliced isoforms were all increased in soybean roots [34]. Transgenic soybean hairy roots and Arabidopsis overexpressing of *GmACBP3* or *GmACBP4* were more sensitive to salt stress compared to wild plants, while transgenic plants overexpressing the alternatively spliced isoforms were more salt-tolerant [34]. In this study, the ankyrin-ACBPs clade contains two *GhACBP* genes, *GhACBP1* and *GhACBP2*, which are induced by salt and drought stresses (Figure 5). GhACBP1 shares 97.55%, 66.19%, 65.83%, 65.74%, and 65.73% sequence identity with GhACBP2, AtACBP1, AtACBP2, GmACBP3, and GmACBP4, respectively. The high similarity of amino acid sequences and expression patterns under abiotic stress indicates that the functions of the ankyrin-ACBPs clade genes are highly conserved. However, further studies are needed to validate the function of *GhACBP1* and *GhACBP2* in abiotic stress tolerance.

## Figures and Tables

**Figure 1 genes-14-00859-f001:**
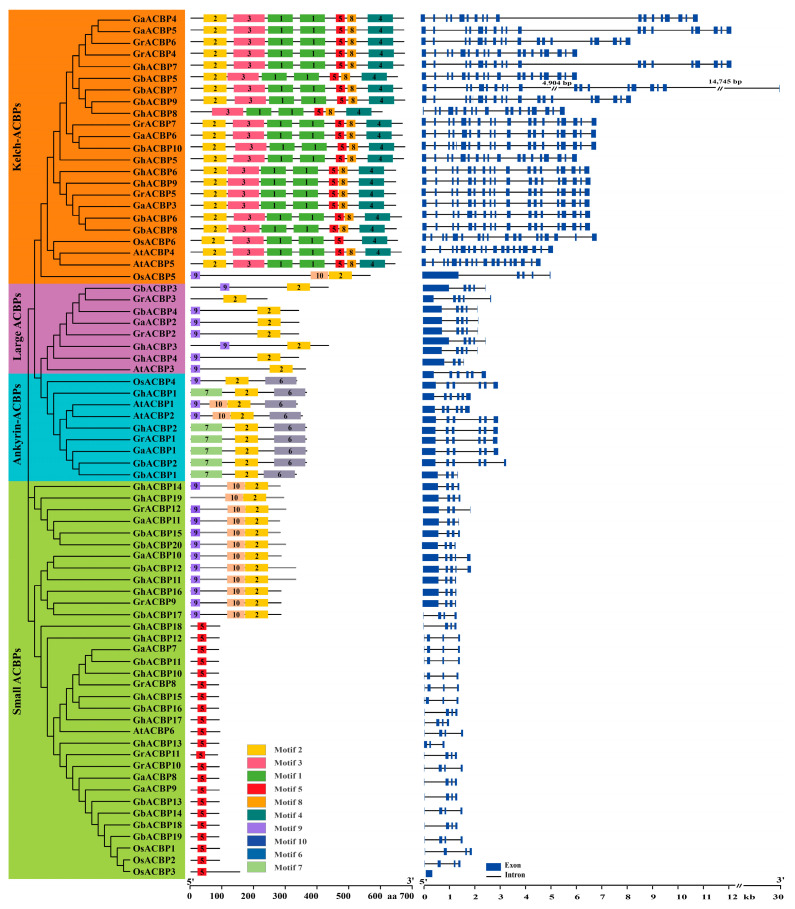
The phylogenetic tree, gene structures, and conserved motifs of cotton *ACBP* genes. The ML phylogenetic tree was developed using MEGA11 with 1000 bootstrap replicates. The boxes with different colors indicate different conserved motifs (Appendix A). The blue boxes and black lines represent exons and introns, respectively.

**Figure 2 genes-14-00859-f002:**
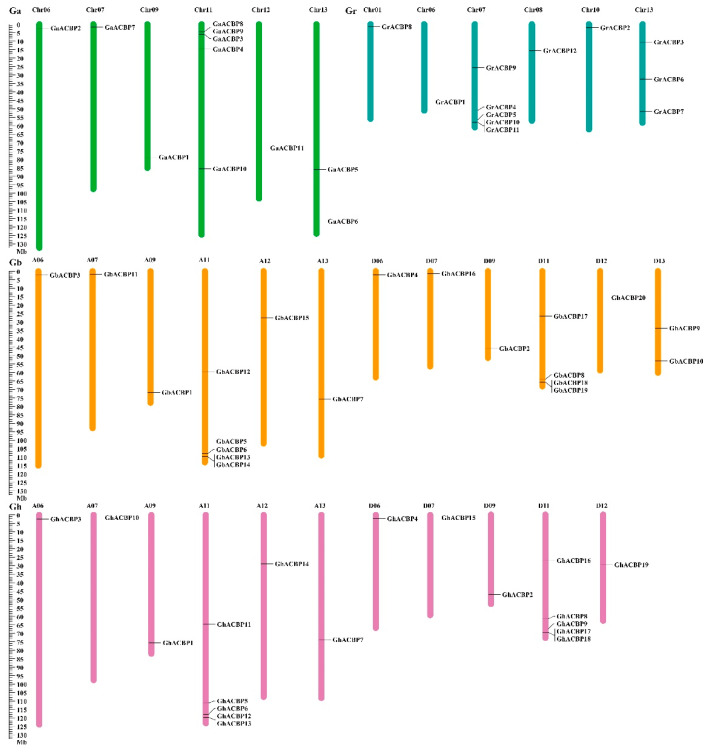
Chromosomal mapping of the *Gossypium ACBP* genes. Bars with different colors denote the chromosomes of four *Gossypium* species, respectively. The scale bar on the left indicates the chromosomal lengths (Mb). Only chromosomes with ACBP genes are shown in the figure.

**Figure 3 genes-14-00859-f003:**
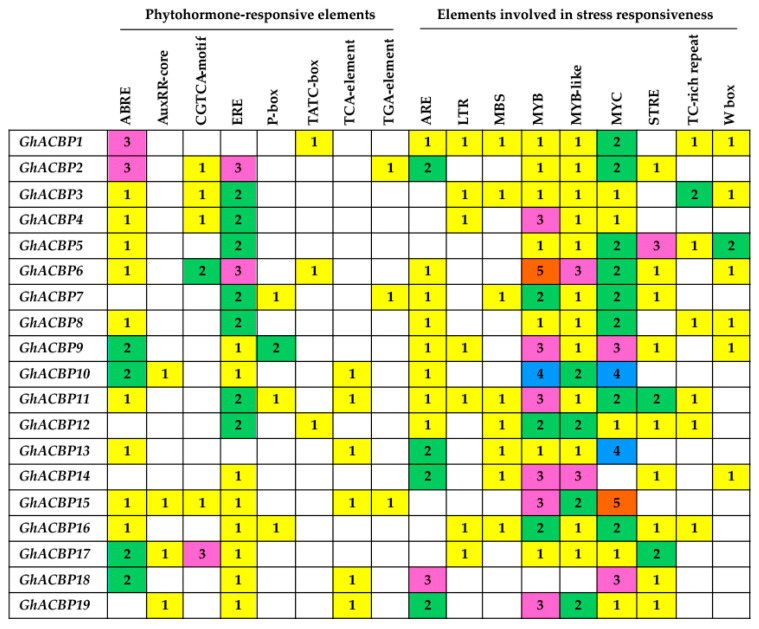
*Cis*-acting regulatory elements in response to phytohormone and stress identified in the promoter regions of *GhACBP* genes.

**Figure 4 genes-14-00859-f004:**
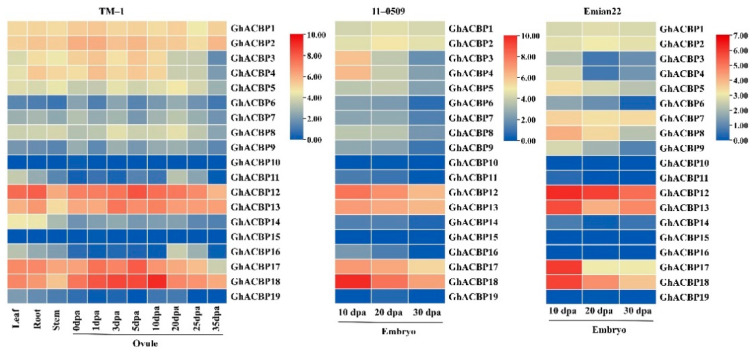
Expression profiles of 19 *GhACBP* genes in different tissues and developmental stages.

**Figure 5 genes-14-00859-f005:**
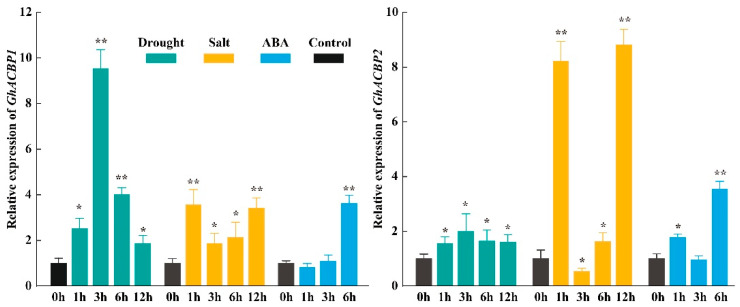
Expression patterns of *GhACBP1* and *GhACBP2* under drought, salt, and ABA treatments determined by RT-qPCR. The standard deviation is indicated by the error bars, and “*” (*t*-test, *p* ≤ 0.05) and “**” (*p* ≤ 0.01) indicate significant differences between the treatment and control.

**Table 1 genes-14-00859-t001:** Identification of *ACBP* genes in four *Gossypium* species.

Gene Name	Gene ID	Chr.	Gene Location (5′-3′)	CDS (bp)	Protein (aa)	MW (kD)	pI	Exon No.
*GaACBP1*	Ga09G2073	Chr09	78,553,212–78,556,181	1107	368	40.08	4.28	6
*GaACBP2*	Ga06G0242	Chr06	2,161,888–2,164,066	1032	343	37.70	4.00	5
*GaACBP3*	Ga11G0450	Chr11	6,033,134–6,039,718	1953	650	71.28	5.31	18
*GaACBP4*	Ga11G0874	Chr11	14,739,861–14,750,764	2028	675	73.66	4.90	18
*GaACBP5*	Ga13G1376	Chr13	85,900,097–85,912,287	2031	676	74.27	5.60	18
*GaACBP6*	Ga13G2190	Chr13	116,634,315–116,641,164	2016	671	73.30	4.86	18
*GaACBP7*	Ga07G0156	Chr07	1,719,913–1,721,334	267	88	9.89	8.86	4
*GaACBP8*	Ga11G0343	Chr11	4,369,004–4,370,524	270	89	10.00	5.89	4
*GaACBP9*	Ga11G0344	Chr11	4,375,942–4,377,237	273	90	10.23	5.21	4
*GaACBP10*	Ga11G1841	Chr11	85,515,304–85,516,624	861	286	31.74	4.34	4
*GaACBP11*	Ga12G2217	Chr12	73,224,127–73,225,994	846	281	31.31	4.04	4
*GrACBP1*	Gorai.006G202200	Chr06	45,862,534–45,866,605	1104	367	39.95	4.25	6
*GrACBP2*	Gorai.010G024700	Chr10	1,965,619–1,968,441	1029	342	37.80	3.98	5
*GrACBP3*	Gorai.013G078800	Chr13	10,499,480–10,502,223	729	242	27.61	9.46	5
*GrACBP4*	Gorai.007G298800	Chr07	50,996,820–51,005,403	2037	678	74.17	4.86	18
*GrACBP5*	Gorai.007G342000	Chr07	56,727,854–56,734,910	1953	650	71.19	4.99	18
*GrACBP6*	Gorai.013G125100	Chr13	32,513,007–32,521,783	2031	676	74.17	5.71	18
*GrACBP7*	Gorai.013G205300	Chr13	51,615,237–51,622,742	2016	671	73.30	4.90	18
*GrACBP8*	Gorai.001G014900	Chr01	1,397,605–1,399,298	267	88	9.86	9.40	4
*GrACBP9*	Gorai.007G221500	Chr07	25,732,138–25,734,387	861	286	31.71	4.22	4
*GrACBP10*	Gorai.007G349800	Chr07	57,989,173–57,990,985	273	90	10.21	5.21	4
*GrACBP11*	Gorai.007G349900	Chr07	57,996,246–5,799,7051	258	85	9.59	5.16	3
*GrACBP12*	Gorai.008G080300	Chr08	15,569,425–15,572,020	906	301	33.51	4.07	4
*GbACBP1*	Gbar_A09G019530	A09	71,712,892–71,716,687	1008	335	36.53	4.36	6
*GbACBP2*	Gbar_D09G019330	D09	45,806,450–45,810,624	1104	367	39.95	4.28	6
*GbACBP3*	Gbar_A06G002170	A06	2,339,193–2,342,347	1311	436	48.83	4.25	5
*GbACBP4*	Gbar_D06G002250	D06	2,169,955–2,172,708	1029	342	37.85	4.02	5
*GbACBP5*	Gbar_A11G028440	A11	100,646,756–100,653,463	1971	656	71.80	4.94	18
*GbACBP6*	Gbar_A11G031740	A11	107,991,123–107,998,178	2010	669	73.45	5.08	18
*GbACBP7*	Gbar_A13G012150	A13	75,687,001–75,714,191	2013	670	73.62	5.70	19
*GbACBP8*	Gbar_D11G032520	D11	64,081,464–64,088,613	1959	652	71.56	5.07	18
*GbACBP9*	Gbar_D13G011800	D13	33,826,642–33,835,279	2040	679	74.61	5.82	19
*GbACBP10*	Gbar_D13G019150	D13	53,310,527–53,318,013	2040	679	74.106	4.712	19
*GbACBP11*	Gbar_A07G001530	A07	1,702,904–1,704,442	267	88	9.89	8.86	4
*GbACBP12*	Gbar_A11G022440	A11	59,361,384–59,363,924	999	332	37.41	4.41	5
*GbACBP13*	Gbar_A11G032690	A11	109,782,889–109,784,801	273	90	10.21	5.18	4
*GbACBP14*	Gbar_A11G032700	A11	109,789,213–109,791,374	270	89	10.02	5.90	4
*GbACBP15*	Gbar_A12G007880	A12	27,624,133–27,626,733	852	283	31.53	4.10	4
*GbACBP16*	Gbar_D07G001530	D07	1,536,362–1,539,288	267	88	9.80	9.48	4
*GbACBP17*	Gbar_D11G021460	D11	26,688,623–26,691,070	861	286	31.75	4.25	4
*GbACBP18*	Gbar_D11G033470	D11	65,738,849–65,740,669	273	90	10.21	5.21	4
*GbACBP19*	Gbar_D11G033480	D11	65,745,816–65,747,314	270	89	10.02	5.90	4
*GbACBP20*	Gbar_D12G007830	D12	15,705,344–15,707,870	903	300	33.51	4.10	4
*GhACBP1*	Ghir_A09G019320	A09	75,596,866–75,600,857	1104	367	40.03	4.30	6
*GhACBP2*	Ghir_D09G018830	D09	47,230,179–47,234,288	1104	367	39.92	4.28	6
*GhACBP3*	Ghir_A06G002230	A06	2,488,919–2,492,044	1314	437	48.89	4.27	5
*GhACBP4*	Ghir_D06G002060	D06	2,200,409–2,203,279	1029	342	37.85	4.00	5
*GhACBP5*	Ghir_A11G028840	A11	111,279,451–111,286,281	2028	675	73.87	5.00	18
*GhACBP6*	Ghir_A11G032190	A11	118,023,421–118,030,387	1953	650	71.26	5.00	18
*GhACBP7*	Ghir_A13G011410	A13	73,709,513–73,722,200	2031	676	74.23	5.59	18
*GhACBP8*	Ghir_D11G029020	D11	61,586,023–61,592,814	1824	607	66.57	4.70	17
*GhACBP9*	Ghir_D11G032680	D11	67,415,479–67,422,462	1953	650	71.26	4.94	18
*GhACBP10*	Ghir_A07G001520	A07	1,613,013–1,614,450	267	88	9.89	8.86	4
*GhACBP11*	Ghir_A11G023070	A11	64,631,311–64,633,846	999	332	37.41	4.41	5
*GhACBP12*	Ghir_A11G033220	A11	119,683,936–119,685,963	273	90	10.21	5.18	4
*GhACBP13*	Ghir_A11G033230	A11	119,690,233–119,692,421	270	89	10.02	5.90	4
*GhACBP14*	Ghir_A12G007910	A12	28,971,057–28,973,672	852	283	31.58	4.10	4
*GhACBP15*	Ghir_D07G001540	D07	1,575,302–1,576,882	267	88	9.86	9.40	4
*GhACBP16*	Ghir_D11G021280	D11	27,187,461–27,190,098	861	286	31.78	4.25	4
*GhACBP17*	Ghir_D11G033830	D11	69,635,385–69,637,499	273	90	10.22	5.21	4
*GhACBP18*	Ghir_D11G033840	D11	69,642,750–69,644,268	279	92	10.54	5.86	4
*GhACBP19*	Ghir_D12G009120	D12	29,686,393–29,688,931	885	294	32.69	4.08	4
*AtACBP1*	AT5G53470	Chr5	21,710,170–21,712,614	1017	338	37.53	4.25	6
*AtACBP2*	AT4G27780	Chr4	13,847,549–13,849,934	1065	354	38.48	4.16	6
*AtACBP3*	AT4G24230	Chr4	12,566,631–12,568,866	1095	364	39.53	3.88	4
*AtACBP4*	AT3G05420	Chr3	1,561,671–1,567,336	2007	668	73.07	4.95	18
*AtACBP5*	AT5G27630	Chr5	9,775,854–9,781,002	1947	648	71.01	6.27	18
*AtACBP6*	AT1G31812	Chr1	11,410,766–11,412,233	279	92	10.39	4.91	4
*OsACBP* *1*	LOC_Os08g06550	Chr8	3,698,312–3,700,553	276	91	10.14	4.87	4
*OsACBP* *2*	LOC_Os06g02490	Chr6	860,905–862,569	276	91	10.25	4.69	4
*OsACBP* *3*	LOC_Os03g37960	Chr3	21,082,861–21,084,238	468	155	17.67	9.06	1
*OsACBP* *4*	LOC_Os04g58550	Chr4	34,810,479–34,813,527	1011	336	35.90	4.23	6
*OsACBP* *5*	LOC_Os03g14000	Chr3	7,591,868–7,597,447	1710	569	61.22	3.99	5
*OsACBP* *6*	LOC_Os03g61930	Chr3	35,105,143–35,112,533	1971	656	71.54	5.05	18

Note: ID, identifier; CDS, coding sequence; MW, molecular weight; aa, amino acid.

**Table 2 genes-14-00859-t002:** Duplicated *ACBP* genes in four *Gossypium* species.

Duplicated Pair	Duplicated Type	Ka	Ks	Ka/Ks
*GhACBP1/GhACBP2*	Segmental	0.011	0.038	0.281
*GhACBP3/GhACBP4*	Segmental	0.021	0.019	1.101
*GhACBP5/GhACBP6*	Segmental	0.109	0.444	0.244
*GhACBP5/GhACBP7*	Segmental	0.080	0.394	0.204
*GhACBP5/GhACBP8*	Segmental	0.011	0.036	0.302
*GhACBP5/GhACBP9*	Segmental	0.110	0.445	0.247
*GhACBP6/GhACBP8*	Segmental	0.110	0.475	0.231
*GhACBP6/GhACBP9*	Segmental	0.017	0.038	0.452
*GhACBP7/GhACBP8*	Segmental	0.080	0.412	0.194
*GhACBP8/GhACBP9*	Segmental	0.112	0.475	0.236
*GhACBP10/GhACBP15*	Segmental	0.025	0.073	0.335
*GhACBP11/GhACBP14*	Segmental	0.238	0.499	0.478
*GhACBP11/GhACBP16*	Segmental	0.050	0.085	0.590
*GhACBP12/GhACBP17*	Segmental	0.019	0.018	1.071
*GhACBP14/GhACBP16*	Segmental	0.202	0.440	0.459
*GhACBP12/GhACBP13*	Tandem	0.102	0.299	0.341
*GhACBP17/GhACBP18*	Tandem	0.219	0.369	0.592
*GbACBP1/GbACBP2*	Segmental	0.046	0.072	0.638
*GbACBP3/GbACBP4*	Segmental	0.019	0.029	0.642
*GbACBP5/GbACBP6*	Segmental	0.105	0.440	0.238
*GbACBP5/GbACBP7*	Segmental	0.077	0.393	0.196
*GbACBP5/GbACBP8*	Segmental	0.108	0.445	0.243
*GbACBP5/GbACBP9*	Segmental	0.077	0.380	0.203
*GbACBP5/GbACBP10*	Segmental	0.079	0.368	0.216
*GbACBP6/GbACBP8*	Segmental	0.016	0.040	0.391
*GbACBP7/GbACBP9*	Segmental	0.013	0.054	0.235
*GbACBP7/GbACBP10*	Segmental	0.085	0.390	0.219
*GbACBP9/GbACBP10*	Segmental	0.087	0.375	0.233
*GbACBP11/GbACBP16*	Segmental	0.025	0.036	0.683
*GbACBP12/GbACBP15*	Segmental	0.240	0.500	0.480
*GbACBP12/GbACBP17*	Segmental	0.050	0.080	0.624
*GbACBP13/GbACBP18*	Segmental	0.014	0.018	0.801
*GbACBP15/GbACBP17*	Segmental	0.204	0.441	0.462
*GbACBP13/GbACBP14*	Tandem	0.102	0.299	0.341
*GbACBP18/GbACBP19*	Tandem	0.107	0.380	0.282
*GaACBP3/GaACBP4*	Segmental	0.107	0.436	0.245
*GaACBP3/GaACBP5*	Segmental	0.119	0.428	0.279
*GaACBP4/GaACBP5*	Segmental	0.081	0.386	0.210
*GaACBP10/GaACBP11*	Segmental	0.217	0.494	0.439
*GaACBP5/GaACBP6*	Segmental	0.083	0.384	0.217
*GaACBP8/GaACBP9*	Tandem	0.096	0.353	0.273
*GrACBP9/GrACBP2*	Segmental	0.457	1.588	0.287
*GrACBP4/GrACBP5*	Segmental	0.110	0.467	0.235
*GrACBP4/GrACBP6*	Segmental	0.075	0.387	0.194
*GrACBP4/GrACBP7*	Segmental	0.078	0.379	0.206
*GrACBP5/GrACBP6*	Segmental	0.116	0.448	0.260
*GrACBP5/GrACBP7*	Segmental	0.116	0.428	0.270
*GrACBP6/GrACBP7*	Segmental	0.083	0.368	0.225
*GrACBP10/GrACBP11*	Tandem	0.113	0.311	0.364

## Data Availability

Not applicable.

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
