# Peer review of "Molecular Characterization of the Acyl-CoA-Binding Protein Genes Reveals Their Significant Roles in Oil Accumulation and Abiotic Stress Response in Cotton"

_genes, 2023, doi:10.3390/genes14040859_

Round 1

Reviewer 1 Report

Please see coments on the manuscript.

Author Response

Highlights: line 22 “abiotic tolerance” and line 58“provide”

Reply: Thanks. Two sentences have been revised.

Notes:

  1. Line 105: What was the design of the experiment?

Reply: Thanks. We have added the experimental design in the revised manuscript.

  1. Line 106: Your manuscript showing crucial role of this gene family in embryo development and seed oil contents. But your ex-plant for expression analysis is only leaves.

Reply: Thanks for the good comments. The genes expression analysis containing embryos or seeds are more scientific. In fact, we have collected a number of cotton genotypes and are determining their seed oil content. In further studies, we will determine the expression patterns of GhACBP genes in cotton embryos and seeds.

  1. Line 142: Figure quality maybe improved, Line 184: Improve the figure quality, Line 247: Clearly indicate the normal treatment other than stress.

Reply: Thank you. The three pictures have been changed and replaced more clear versions.

  1. Line 282: The oil contents maybe tested in seeds to correlate with gene expression analysis.

Reply: Thanks for your valuable comments. The ACBP genes play a crucial role in embryo development and seed oil content. In maize, the expression levels of five ZmACBP genes were directly correlated with maize kernel oil content. In further studies, we will test the correlation of expression levels of GhACBP genes with cotton seed oil content.

Reviewer 2 Report

Chen et al performed molecular characterization of ACBP genes in cotton. They found 11 and 12 copies of ACBP genes in diploid progenitors Ga and Gr, respectively. While two allotetraploid progenies, Gb and Gh, have 20 and 19 ACBP genes. The ACBP genes in these two species are nearly doubled. They also found that most of these genes have undergone purifying selection. Most of the results from the manuscript make sense. I have some minor questions and suggestions. 

Line 16. It might be good to mention the relationship between these four cotton species in the Introduction. I'd like to point out that readers out of the cotton area need such information to grab your points quickly. You don’t want them to go to the Discussion part to find GA and GR are diploid progenitors of two tetraploid progeny species Gb and Gh. 

Line 141, Figure 1. I always recommend authors add bootstrap values to their phylogenic tree. If you have a bootstrap value of 200 in a clade instead of 900 out of 1000 bootstrap replicates, that clade should be less confident.  

Line 184, You might need to add “Only chromosomes with ACBP genes shown in the figure” in the legend of Figure 2. 

There is no legend for supplemental figures.

Author Response

Chen et al performed molecular characterization of ACBP genes in cotton. They found 11 and 12 copies of ACBP genes in diploid progenitors Ga and Gr, respectively. While two allotetraploid progenies, Gb and Gh, have 20 and 19 ACBP genes. The ACBP genes in these two species are nearly doubled. They also found that most of these genes have undergone purifying selection. Most of the results from the manuscript make sense.

Thanks for the good comments on our manuscript.

  1. Line 16: It might be good to mention the relationship between these four cotton species in the Introduction. I'd like to point out that readers out of the cotton area need such information to grab your points quickly. You don’t want them to go to the Discussion part to find GA and GR are diploid progenitors of two tetraploid progeny species Gb and Gh.

Reply: Thanks for your comments. The contents of the relationship between four cotton species have been added in the introduction (Page 3).

  1. Line 141: Figure 1. I always recommend authors add bootstrap values to their phylogenic tree. If you have a bootstrap value of 200 in a clade instead of 900 out of 1000 bootstrap replicates, that clade should be less confident.

Reply: Thanks for your comments. ML phylogenetic tree will be more visual with bootstrap values. We have added a ML phylogenetic tree in the Supplementary Materials with bootstrap values higher than 30 (Figure S1).

  1. Line 184: You might need to add “Only chromosomes with ACBP genes shown in the figure” in the legend of Figure 2.

Reply: Thanks. The legend of Figure 2 has been revised.

  1. There is no legend for supplemental figures.

Reply: Thanks. The legends for supplemental figures have been added in the Supplementary Materials of Page 12.

Round 2

Reviewer 1 Report

The manuscript is fine now.